# Effects of Cattle Traffic on *Sclerocactus wrightiae*

David Lariviere [1,*], Val Anderson [1], Robert Johnson [2], Tyson Terry [3] and Thomas Bates [1]

1 Department of Plant and Wildlife Sciences, Brigham Young University, 4105 LSB, Provo, UT 84602, USA
2 Department of Biology, Brigham Young University, 4102 LSB, Provo, UT 84602, USA
3 Department of Wildland Resources, Utah State University, 5230 Old Main Hill, NR 206, Logan, UT 84322, USA
* Correspondence: david.d.lariviere@gmail.com

**Abstract:** Cattle grazing has been a historic use of rangelands in Utah since pioneer settlement in the mid-1800's. Wright fishhook cactus is a small globose cactus endemic to an area of 280,000 ha in south–central Utah and was listed as endangered in October of 1979, by the U.S. Fish and Wildlife Service (USFWS). By 2010, concerns were expressed that soil compaction in proximity to the cactus posed a threat to this species, though there were no empirical data to support such concerns. In order to assess the impact of cattle traffic on Wright fishhook cactus, we used an imprint device to simulate a cow track's impact. We applied a treatment of either zero, one, or four hoof imprints within 15 cm evenly of 146 cacti within the same population cluster on the same day. We monitored subsequent plant survival as well as reproductive success. Each cactus in the study was visited multiple times and all developed seed was collected. We found that cattle traffic of any amount had no effect on plant survival or seed production and, therefore, concluded that cattle traffic poses no threat to Wright fishhook cactus. The status of this cactus yields no justification for changing the historic land management use of cattle grazing on these rangelands.

**Keywords:** cacti; cattle impacts; desert ecology; livestock effects; reproductive fitness; seed production; range management; endangered cacti; land management; globose cactus

## 1. Introduction

Domesticated cattle (*Bos taurus*) have been present throughout the western United States since settlement in the early 19th century, and in large numbers on Utah rangeland since approximately 1850 [1]. The presence of cattle was largely detrimental to native plant communities in early history, but this negative impact has been drastically reduced with modern management practices [2]. Cattle can also have positive effects on certain native species when managed properly [3,4]. Cattle can boost overall productivity of native perennial bunch grasses by increasing light penetration through the elimination of superfluous plant material [3]. Wavy-leaf thelypody (*Thelypodium repandum*) has been shown to germinate and establish at dramatically higher rates in the presence of cattle grazing [4]. When properly managed, cattle are removed from sites before they can overgraze and downgrade the native plant community. However, the effects of cattle are not limited to grazing. Cattle also step, defecate, urinate, and bed on plants that they may not have otherwise affected by grazing [5].

Cattle traffic can have a large impact on soil compaction and erosion. In riparian zones, and the presence of cattle has led to increases in stream turbidity and rates of soil erosion [6]. The long-term presence of cattle has been shown to lead to soil compaction in pasturelands [7,8]. This effect may extend to more than just soil properties. When paired with long-term defoliation, cattle traffic has been shown to influence species composition in native grasslands [9]. It is this effect that is of interest here. Currently, there is controversy surrounding the presence of cattle in the San Rafael Desert region of east–central Utah. Several federally listed plant species exist in this region, but the most widespread is the

endangered Wright fishhook cactus (*Sclerocactus wrightiae*). This study examines the impact of cattle traffic on this species.

Wright fishhook cactus is a small globose cactus endemic to three counties in south–central Utah (Figure 1) [10,11]. Wright fishhook cactus has the ability to retract into the soil during dormancy, emerging with sufficient rainfall to bloom in late April to early May, making them uniquely difficult to inventory in drought years [12]. Wright fishhook cactus was not distinguished from the much more common little-flower fishhook cactus (*Sclerocactus parviflorus*) until its publication in 1966 [10]. It is differentiated by its white flowers and magenta filaments. In 1979, Wright Fishhook cactus was listed as endangered by the U.S. Fish and Wildlife Service (USFWS) [13], citing its very limited range, population size, and the prevalence of poaching by international cactus hobbyists [13]. Since the time of its listing, impacts from cattle have also been identified as an existential threat to the species [13–16].

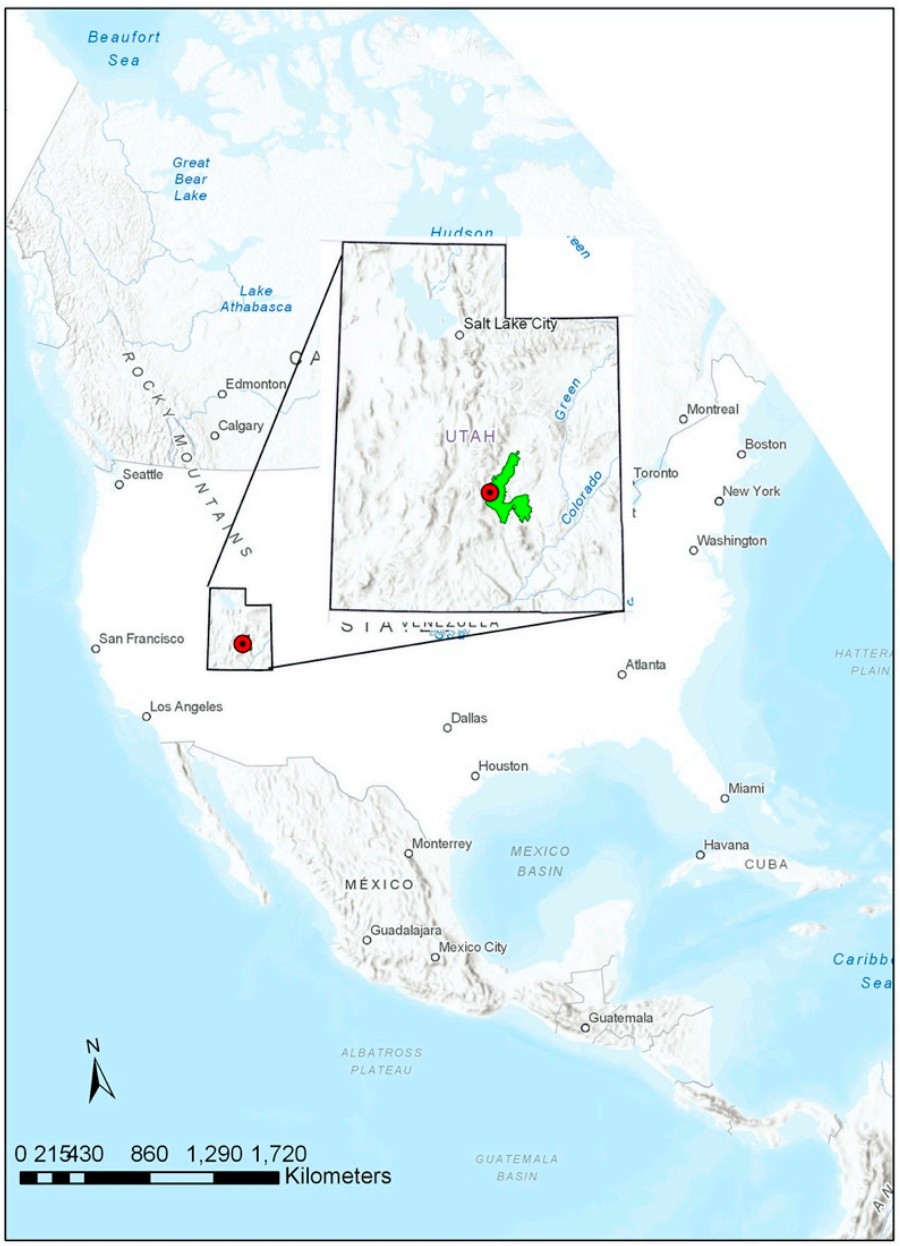

**Figure 1.** (Range map of *Sclerocactus wrightiae*, red point is the study site. Centroid of range: 38.5318460, −111.2391515).

The effects of cattle on endangered plants have been a management concern since the late 1980's [17]. Certain groups within cactaceae have been shown to be particularly susceptible to this potential threat [18]. However, the response of all cacti to the presence of cattle is not consistent and cannot be generalized. Saguaro (*Carnegiea gigantea*) and a species of pincushion cactus (*Mammillaria dixanthocentron*) have been shown to decrease in population size in the presence of cattle [19,20]. However, that globose cacti favor the disturbed conditions caused by the presence of well-managed cattle is well-documented [19,21–24]. An endangered pincushion cactus (*Mammillaria hernandezii*) favored the disturbed conditions caused by cattle, with their population growth rate increasing in the presence of cattle grazing [19], while another endangered globose cactus, *Mammillaria pectinifera*, increased in population density under moderate grazing regimes [21]. The response to grazing and disturbance appears species-specific.

Under the assumption that the presence of cattle had the potential to reduce Wright Fishhook cactus populations and functional range, an interagency team was created between the Bureau of Land Management (BLM), Capitol Reef National Park (CRNP), and the USFWS to monitor the impacts and disturbances created by cattle. Major concerns for the species were that cattle may directly harm the cactus by stepping on them, uprooting them, shearing their roots, burying them, or indirectly harming individuals through soil compaction, reduced water availability, and/or reduced nutrient availability [14,25,26].

In 2011, a committee with members from the USFWS, National Park Service (NPS), and BLM arbitrarily outlined cattle as having disturbed the cactus when a hoofprint is found within 15 cm of the cactus base, despite no empirical evidence indicating that a cattle print at this distance had deleterious effects on this species. This number was based on the 15 cm average length for shallow horizontal roots of the closely related Uinta Basin hookless cactus (*Sclerocactus glaucus*) [10] and the average diameter of an adult cow hoof, which is 10 cm [14,27]. An interagency team was created and instructed to record the presence of any cattle tracks within the defined 15 cm radius of any *Sclerocactus wrightiae* they encountered during their inventory. If cattle prints were found within this distance on 15% or more of the cacti within a key area (defined as groupings of cactus locations within distinct geographic areas of their range) then that area would need to be reviewed by the USFWS and, subsequently, have its cattle grazing permits reduced by the BLM until fewer than 15% of cacti were found to have cattle hoofprints within 15 cm of their base [28].

After the formation of this interagency agreement, both Capitol Reef National Park and the BLM (the two land management agencies primarily responsible for the cacti's range) devoted much time and effort into monitoring the species. From 2011 to 2013, the BLM collected observational data from 8767 individual Wright fishhook cacti, examining diameter, number of stems, and reproductive effort, finding that there was no significant correlation between population density and cattle tracks within 15 cm of the cactus base [29]. A follow-up study using BLM data through 2017 [30] found no correlation between the levels of cattle disturbance and changes in cacti population or reproductive structure densities per macro plot. Capitol Reef National Park began their own observational study from 2013–2016, which looked at 352 individual cacti and found that through impacts to the community structure at large, Wright Fishhook cactus populations were indirectly negatively impacted by the presence of cattle [31]. Since the completion of this second observational study, proper management of the species has been a point of disagreement for the BLM, USFWS, and CRNP. With the 2013/2022 BLM study and 2017 National Park study being observational, the need for a treatment study measuring the effects of cattle traffic on Wright fishhook cactus was apparent in order to reach an inter-agency consensus on management. In 2018, Brigham Young University initiated a study on Wright fishhook cactus to determine experimentally the effects of cattle traffic on Wright fishhook cactus. Our study offers a controlled traffic treatment to a large, uniform population of Wright fishhook cactus in order to analyze the impacts of a cow stepping in close proximity to a cactus-on-cactus reproductive fitness. Despite observation occurring across only two years, our study served to test the assumption of the existing land management policy, which

defines cacti as having been immediately disturbed by a single cattle hoofprint within 15 cm of their base.

## 2. Materials and Methods

### 2.1. Site Description

The study site is located on 4 hectares of arid desert that is privately owned and heavily grazed. It is approximately 14 km south of Fremont Junction, Utah (latitude 38°63′ N, longitude 111°33′ W) in what is known as the Last Chance Wash. The red point in Figure 1 indicates the location. The study site was fenced the year prior to the study to prevent the present cattle from confounding our treatment. The region has an arid climate, with an average annual precipitation of 190 mm [32]. The soil at the site is sandy clay loam in texture and is underlain by alluvium [33]. The dominant native plant species in this cold desert scrub community are: shadscale (*Atriplex confertifolia*), sand buckwheat (*Eriogonum leptocladon*), alkali sacaton (*Sporobolus airoides*), galleta (*Hilaria jamesii*), Torrey's ephedra (*Ephedra torreyana*), four-wing saltbush (*Atriplex canescens*), Indian rice grass (*Achnatherum hymenoides*), and prickly pears (*Opuntia* sp.). Non-native species frequently found in this community include Russian thistle (*Salsola tragus*) and halogeton (*Halogeton glomeratus*).

### 2.2. Cow Hoof-Imprint Device

Simulating the effects of cattle traffic has been accomplished by diverse means [34–37]. However, none of these methods were deemed suitably practical for this application, so a novel design was constructed (Figures 2 and 3). The traffic device we constructed for this study is primarily constructed of steel and makes use of two primary components, a pressure platform, and a plunger assembly. The pressure platform measures 55 cm × 55 cm and is 32.5 cm tall. It has four central bars that form an octothorpe around a central guide pipe. The guide pipe is 21.6 cm tall and 11.4 cm in diameter. It is welded in place with 6.4 cm exposed above the plane of the stabilizing platform. The plunger consists of an upper platform measuring 30 cm × 49.5 cm, a flanged pipe fitting measuring 23 cm in diameter, and a PVC pipe 10.2 cm in diameter and 38 cm in height. The plunger assembly slides freely through the guide pipe of the stabilizing platform. A cow leg with a hoof surface area of 82 cm$^2$ was then obtained from a local butcher and mounted onto the bottom of the plunger. The plunger assembly (including leg) weighs 6.8 kg. To adequately simulate the proper pressure applied by a 400 kg cow while walking, two persons with a combined mass of 200 kg would stand simultaneously on the upper platform exerting a total force of 2.5 kg cm$^{-2}$ (245 kPa) (Figure 3).

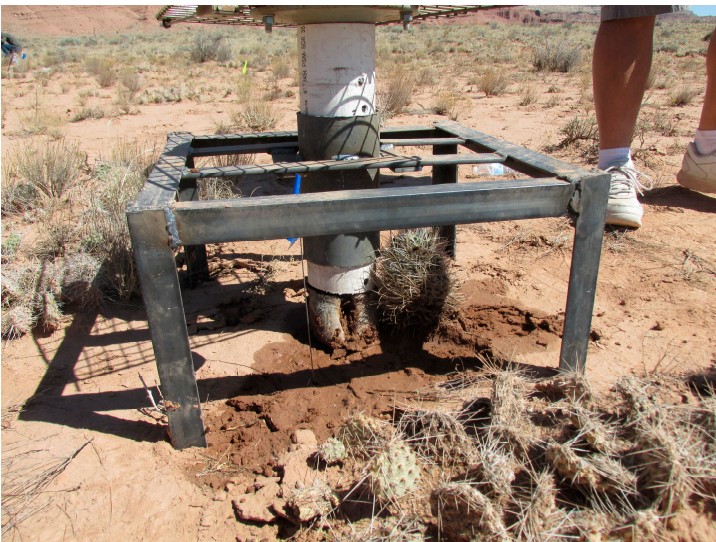

**Figure 2.** Design of imprint device.

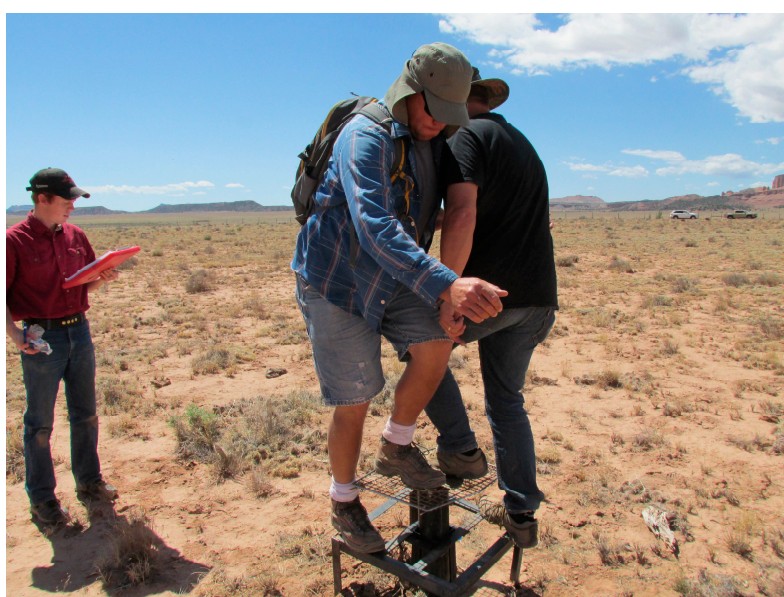

**Figure 3.** Imprint device in use.

### 2.3. Sample Methodology

Four size classes for Wright fishhook cactus have been identified, three of which have flowering capability: class two (cacti 2.1–4 cm in diameter), class three (cacti 4.1–9 cm in diameter), and class four (cacti >9 cm in diameter) [25]. Our study focused on size classes two and three, which are primarily responsible for the reproductive output in our study population, as size class four was not present in our study site and is quite rare across the landscape [25].

The two years of our study had quite different rainfall regimes. In the 2018 treatment year, nearly all the year's precipitation occurred in August (after our treatment and seed collection) in the form of monsoonal rain. Due to a significant early season drought in the 2018 treatment year, many cacti were still subterranean, having received insufficient moisture to develop the necessary turgor to emerge. Only 64 cacti (22 size class two and 42 size class three) were found for inclusion in the study. In 2020, the spring was cool with smaller, more frequent rainfall events throughout the season occurring prior to treatment and seed collection. In this more favorable water year, 146 cacti (69 size class two and 77 size class three) were found and included in the study.

For both treatment years, cattle tracks, our study's treatment, were imposed immediately off the cactus perimeter, with no part of the track reaching 15 cm beyond the base of the cactus. Treatments were, control, one track, and four tracks. Cattle tracks were positioned randomly using a number generator (1, 2, 3, 4) based ordinally in the four cardinal directions (N, S, E, W). Quantity of track treatments were then randomly assigned, maintaining an even proportion of size classes within each treatment.

Each year we monitored reproductive activity over the weeks leading up to the maturation of seeds so as to collect at peak season. We counted the number of flowers produced by each cactus. We also counted aborted flowers that did not result in seed production, and we harvested every seed pod from every cactus. We then counted the number of seeds in each pod using a Data Technologies S-JR laser seed-counting machine.

### 2.4. Analysis

Statistical analyses were performed using R software. Reproductive fitness was examined through three variables, seed quantity (seeds), mature flower quantity (flowers), and number of flowers which were aborted before producing viable seeds (aborted). These data were independently compared against a control group and stratified by two size classes using a negative binomial regression. We chose this modeling approach because

our response variables were in counts and had a zero-inflated distribution. All our models incorporated an intercept term ($\alpha$), fixed effects for hoofprint count ($\beta_1$) and size class ($\beta_2$), with a random effect of year (Year) to account for variability in interannual conditions that may affect seed and flower production. We used this approach to understand effects of hoofprints on seed count (Equation (1)) and flower production (Equation (2)). We used a binomial logistic regression to understand how hoofprints affect the probability of flower abortion (Equation (3)) using the same fixed and random effects as the previous two models. The effect of trampling on cactus mortality was also studied. No cacti of any treatment group perished during the course of our study.

$$\text{Seeds} = \alpha + \beta_1 * \text{Hoofprint Count} + \beta_2 * \text{Size Class} + \text{Year} + \text{error} \qquad (1)$$

$$\text{Flowers} = \alpha + \beta_1 * \text{Hoofprint Count} + \beta_2 * \text{Size Class} + \text{Year} + \text{error} \qquad (2)$$

$$\ln\left[\frac{\text{aborted}}{1 - \text{aborted}}\right] = \alpha + \beta_1 * \text{Hoofprint Count} + \beta_2 * \text{Size Class} + \text{Year} + \text{error} \qquad (3)$$

These models were also validated with a chi-square test against a null model, with residuals checked for homogeneity. We corrected for differences in variation by utilizing a negative binomial distribution to account for overdispersion and disparate levels of variance among high distribution values. Model fit was assessed using pseudo R-squared values, and were found to be 0.37 for Equation (1) and 0.52 for Equation (2).

## 3. Results

Data from the two treatment years were combined for temporal replication and increased sample size due to the treatment effect having no significant difference across years (Figures 4 and 5). In our model, variation between years was treated as a fixed effect. When comparing AIC values between potential models, we found that lumping treatment responses across both years provided the most predictive power. We used a negative binomial distribution for count data (checked against Poisson and quasi-Poisson using AIC values). We then validated our model with a chi-Square test run against a null model and checked residuals for homogeneity. We found the pseudo R-squared value to be 0.33 (log-normal based).

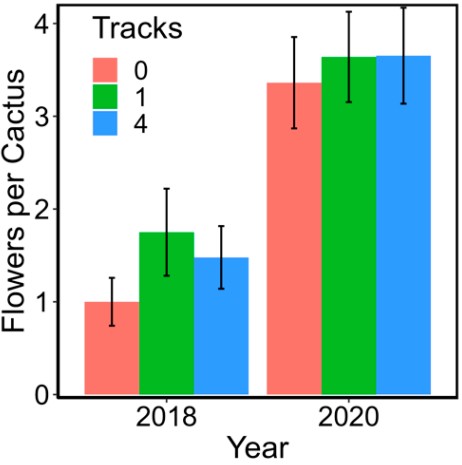

**Figure 4.** Flower production by year and treatment.

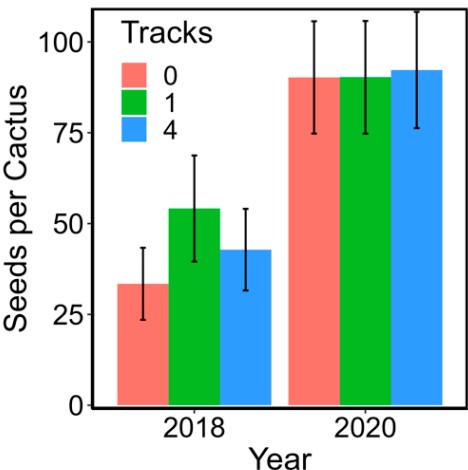

**Figure 5.** Seed production by year and treatment.

## 3.1. Flower Production

We found that cattle traffic had no effect on the number of flowers produced by a given cactus ($p > 0.57$). Over the two years of study, 82 cacti produced no flowers, while 12 cacti produced over ten (Figure 6). There was a significant relationship between size class and flower quantity ($p < 0.0001$) (Figure 7).

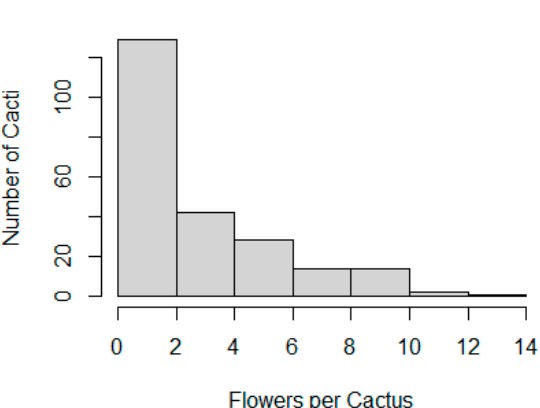

**Figure 6.** Cacti quantity by flowers per cactus.

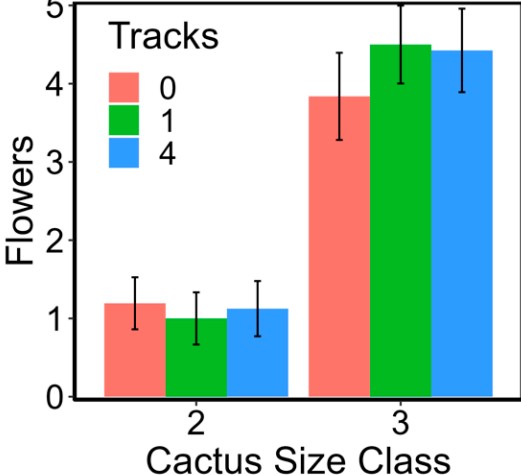

**Figure 7.** Flower production by treatment groups.

### 3.2. Flower Abortion

We found that cattle traffic did not significantly affect whether cacti aborted flowers prior to the production of mature, viable seeds ($p > 0.52$). Over the two years of study, 34 individual cacti aborted at least one flower, while 186 did not abort any. Percentage likelihood of abortion was treated as the response variable in this model. We found that flower abortion increased significantly in larger size classes ($p = 0.012$). Flower abortion rate was also heavily tied to year ($p = 0.008$).

### 3.3. Seed Production

Seed production per individual range from 0 to 506 (Figure 8), with larger size class cacti having a greater variability in seed quantity than those in smaller size classes. The number of cattle tracks did not significantly affect the number of seeds produced by Wright fishhook cactus ($p > 0.92$) (Figure 9). All size classes produced significantly fewer seeds in the 2018 treatment year than in 2020, while plants in the larger size classes produced significantly more seeds than smaller individuals across both years ($p < 0.0001$).

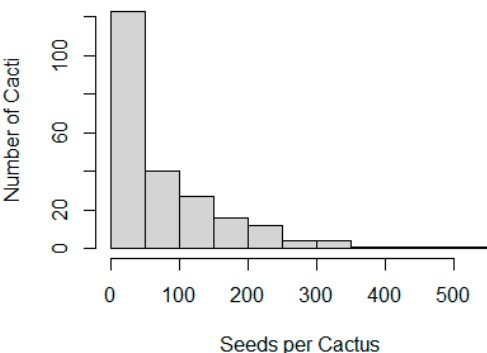

**Figure 8.** Cacti quantity by seeds per cactus.

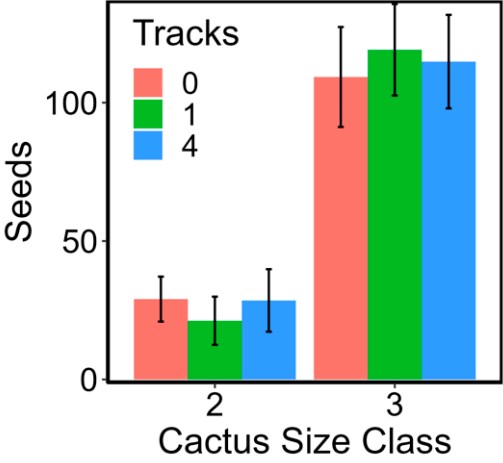

**Figure 9.** Seed production by treatment groups.

## 4. Discussion

Since the USFWS 1985 recovery plan was drafted for this species, publications and some land managers have supposed, without empirical data, the idea that cattle represent a major threat to the survival of Wright fishhook cactus and have considered them a likely contributing factor to cactus population decline [14,15,25,32]. However, the presence of cattle has been met with varying responses from other members of Cactaceae. While populations of saguaro (*Carnegiea gigantea*) have been shown to decline in the presence of cattle [20], it is not clear the cattle behaviors that caused these populations to decline. Cattle

may have used these tall, columnar cactus as rubbing sites, loitered around them for shade, or for some other behavior, which led to increased mortality and decreased recruitment. A vulnerable species of pincushion cactus (*Mammillaria dixanthocentron*) has been shown to decrease in population growth rate when their habitat is overgrazed to the point of massive soil degradation [19]. Appropriate grazing practices should be expected over the range of Wright fishhook cactus, as the vast majority of the species is located on land managed by Federal Agencies. The fact that moderate disturbance caused by the presence of cattle is beneficial to numerous globose cacti species is well-documented [19–24]. Cattle grazing at sustainable levels has been shown to boost seedling recruitment in the endangered pincushion cactus (*Mammillaria hernadezii*), where increased population growth rates were observed under moderately disturbed conditions associated with cattle grazing [19]. Additionally, severity of cattle disturbance has not been found to have a negative effect on the cacti's population density in Wright fishhook cactus [30]. These beneficial effects could potentially be due to prolonged evolution of globose cacti in the presence of now-extinct American megafaunal species over the past two million years [38].

We found traffic to have no significant effect on the reproductive fitness of Wright fishhook cactus when tracks weighted with 200 kg fell within 15 cm of cacti bases. This finding would imply that there is no danger posed to the reproductive fitness of these cacti when cattle are permitted to move through their habitat. Even in the unlikely event of four steps (one on each cardinal side) within 15 cm of the base of these cacti, the number of seeds, number of flowers, and rate of flower abortion were unaffected.

Our finding, as well as that of Bates et al. [30], contradicts that of Hornbeck [31]. The conclusions of that report cited effects at the community level in the presence of cattle in order to assess consequences to the fitness of Wright fishhook cactus individuals. Given our finding that cactus reproduction, as measured in the number of flowers and seed output, is not negatively impacted by even four track treatments within 15 cm of the base of the cactus, it seems unlikely that the general effects of cattle presence, as cited by Hornbeck [31], create measurable impacts on populations of this species. This is supported by the results of the large-sample, multi-year observational study published by Bates et al. in 2022 [30], which found that the severity of cattle disturbance had no impact on Wright fishhook cactus population density over time.

Due to the branched taproot structure of this cactus, the absence of any discernible impact on the cactus' reproductive output by cattle traffic should be expected. The Wright fishhook cactus does not have sprawling horizontal root structures and does not primarily absorb water at shallow soil depths [10]. Any soil compaction caused by the cattle hoof appears to have little effect on the cactus root structure and water-gathering capabilities. Additionally, because this species lacks a significant shallow horizontal root structure, there is little chance that cattle tracks near the base of these cacti would lead to significant root shearing.

Notably, our data indicate that, taken together, size class and year explain 90% of the observed variation in flower production. Any study looking at reproductive output trends of this species over multiple years should consider these two variables in their analysis, especially when beginning or concluding observation on an above or below average rainfall year. Such a response by the cacti to environmental variation may skew results in an observational study to indicate trends that would be unrelated to other variables.

Post-dispersal seed germination rates were not examined in this study. The effect cattle hoof prints have on the alteration of the micro-habitat surrounding cactus has not been studied and is not currently known. The presence of high-quality micro-habitats has been shown to significantly increase germination rates among cacti [39,40]. The slight shading of cacti seeds during early stages of germination in arid environments has been shown to increase recruitment rates [41]. Cattle hoof prints and/or dung may prove to provide excellent micro-habitats for cacti seeds by increasing availability to moisture and providing moderate shading, as ungulate hoofprints have been shown to provide for other species [42]. The authors of this paper have personally observed numerous Wright fishhook cactus germinating in cattle hoof-prints. Cattle hoofprints have been shown to increase

seedbank retention and germination rates in a wide variety of species when seed runoff was of primary concern [43–45]. It is not known what effect the annual, intense monsoonal rainfall this region receives has on seed runoff and loss for this species. There may also exist unknown, existing micro-habitats that cattle dung and hoofprints destroy.

Moving forward, future recovery plans should amend the notion that the presence of cattle negatively impacts Wright fishhook cactus, as our data do not support this assumption. These effects have been shown to be non-existent both in this study and by the work of Bates et al. [30].

**Author Contributions:** Conceptualization, D.L., V.A., R.J. and T.B.; methodology, D.L., V.A. and R.J.; software, D.L. and T.T.; validation, T.T.; formal analysis, T.T.; investigation, D.L., R.J. and T.B.; resources, D.L., V.A. and R.J.; data curation, D.L.; writing—original draft preparation, D.L.; writing—review and editing, V.A. and R.J.; visualization, D.L. and T.T.; supervision, V.A.; project administration, V.A.; funding, V.A. All authors have read and agreed to the published version of the manuscript.

**Funding:** This research was funded internally by Brigham Young University.

**Ethical Statement:** No voucher specimens were collected of this endangered species (*Sclerocactus wrightiae*) in order to preserve the study population.

**Data Availability Statement:** The data presented in this study are publicly available in figshare at https://doi.org/10.6084/m9.figshare.22192930.

**Conflicts of Interest:** The authors declare no conflict of interest. The funders had no role in the design of the study; in the collection, analyses, or interpretation of data; in the writing of the manuscript; or in the decision to publish the results.

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
