# Peer review of "Effects of Cattle Traffic on Sclerocactus wrightiae"

_land, doi:10.3390/land12040853_

Round 1
Reviewer 1 Report
The manuscript by Lariviere presents an experiment of cattle traffic impact on an endangered plant species. It is well written and concise and I think it deserves publication. Still, I have the following minor comments to be addressed before it can be accepted for publication.
l. 36 Please put Latin names in italics
l. 50 Please be consistent whether you use authority names of species or not
Figure 1 Please provide either a coordinate grid or the coordinate of the centroid of the range
l. 197-200 What does the symbol # mean? If it means number, please omit it because it is not necessary. What does “YN” mean? There is something I do not understand of the formulation of these three models. How can be Year introduced as a parameter without any coefficient? I assume Year is a random factor, but I am afraid the authors must rewrite the formulas with a better symbology. In any case, the formulas used must be better explained and reported.
Good luck with the revisions.
Author Response
- Italicize Latin names- Thank you for noticing this, I have corrected the font.
- Use of authority names- Again, thank you for noticing this issue. I have omitted the authority name for this species in order to be consistent in format.
- Figure 1- I have updated the description to include the coordinate of the centroid
- I have updated the equations and their methodology description. Additionally, I have added another paragraph explaining our methodology in greater detail. # did refer to number, so it was omitted. YN was in reference to yes or no. I had mistakenly omitted the coefficient from the formulas. The formulas have been rewritten for accuracy and clarity.
Thank you for your review
Reviewer 2 Report
Comments on MS land-2286770
This is research that meets practical information needs.
I'd like to see a photo of this imprint device.
It is not surprising that the simulated cow track impact had not impact on plant survival. It is likely that the plants have adapted during evolution to the presence of herbivores on the prairies.
Author Response
- I have included a photo of the imprint device as “photo 1”
- Thank you for bringing this point up, I elected to include a reference to it in the discussion, lines 276-278 and reference #38. However, the ecosystem is arid desert rather than American prairie. I added verbiage to clarify this point in the site description lines 128-129.
Reviewer 3 Report
Review of manuscript land-2286770 “Effects of Cattle Traffic on Sclerocactus wrightiae.”
General comment
This paper analyzes the impact of cattle activity on Wright’s fishhook cactus (Sclerocactus wrightiae) —a small globose cactus endemic to south-central Utah and listed as endangered— through field experiments simulating cattle track impact. The study found that cattle traffic had no effect on plant survival or seed production, and concluded that cattle traffic poses no threat to the cactus and that there is no experimental justification for changing the historic cattle grazing regime in these rangelands.
The study seems carefully done, the statistics are adequate, and the results are of ecological and practical importance. I only have a minor comment, related to the background and context of this study in relation to the ecology of these cacti.
The study starts asserting that Sclerocactus wrightiae is a barrel cactus, and the authors analyze their results within that perspective. I disagree. The usual functional-morphological classification of cactoid plants reserves the description of barrel cacti to large, rounded (but not columnar) cacti of North America in the genera Echinocactus and Ferocactus. From the point of view of their functional morphology, Wright’s fishhook cactus (Sclerocactus wrightiae) is considered a globose species, i.e., small cacti that grow close to the ground and are usually dome-shaped, not barrel-like.
Globose cacti occur basically in two tribes of the subfamily Cactoideae within the family Cactaceae. In North America, the globose growth form is common in the Tribe Cacteae, and includes genera such as Ariocarpus, Coryphantha, Escobaria, Geohintonia, Lophophora, Mammillaria, Obregonia, Pelecyphora, and Sclerocactus, among others. In South America, globose growth commonly appears in the Tribe Trichocereeae, and includes genera such as Acanthocalycium, Aylostera, Echinopsis, Gymnocalycium, Matucana, Mila, Oroya, and Rebutia, among others.
The functional distinction between both growth forms is important because it has been frequently reported in the literature that globose growth forms of different genera of cacti can withstand trampling quite effectively. For example, in central-southern Mexico Peters and Martorell (2005), as well as Ureta and Martorell (2009, cited in this manuscript), and have described how some Mammilaria cacti perform optimally under intermediate-intensity grazing. Similarly, in the semiarid ranges of central Argentina, Cingolani et al. (2014) and Lorenzati et al. (2022) have shown that different species of the globose genus Gymnocalycium perform optimally under intermediate grazing intensity and some fire disturbance.
In particular, Cingolani et al. (2014) discuss how the pasture drylands of the Americas were continuously occupied by large herbivores for long geological times, and certainly during the last 2 million years, during which they were trampled and grazed by the now extinct Pleistocene megafauna. Even after the extinction of Pleistocene mega-herbivores about 10 000 years ago, these ecosystems sustained populations of wild middle-sized herbivores, such as bison and deer in North America, and camelids, deer and giant rodents in South America, until their population decrease or local extirpation with the arrival of European settlers.
In short, there is good evidence that Sclerocactus wrightiae, like many other dome-shaped globose cacti in the semiarid grasslands of the Americas has been evolving under moderate levels of grazing pressure for the last two million years, and not only since the introduction of European cattle, 400 years ago. I believe that this fact, as well as the many other studies that have shown globose cacti to perform well under moderate grazing pressure, should be reviewed and discussed in more detail in this study.
Specific comments
Line 320: “Sclerocactus wrightiae plants were treated in this study.” The meaning and purpose of this sentence is unclear. Is there a “No” missing at the beginning of the sentence? Please clarify
Minor comments and edits throughout the manuscript inserted directly in the pdf file, attached.
References
Cingolani, A.M., et al. 2014. Can livestock grazing maintain landscape diversity and stability in an ecosystem that evolved with wild herbivores? Perspectives in Plant Ecology, Evolution and Systematics 16 (2014) 143–153.
Lorenzati, M.A., et al. 2022. Do fires affects growth, seed production and germination of the globose cactus Gymnocalycium monvillei? Journal of Arid Environments 197 (2022), 104663.
Peters, E., and C. Martorell. 2005. The measurement of chronic disturbance and its effects on the threatened cactus Mammillaria pectinifera. Biological Conservation 124 (2005): 199-207
Ureta, C., Martorell, C., 2009. Identifying the impacts of chronic anthropogenic disturbance on two threatened cacti to provide guidelines for population-dynamics restoration. Biol. Conserv. 142, 1992–2001. https://doi.org/10.1016/j.biocon.2008.12.031

Author Response
- Thank you for bringing the classification issue to our attention. I have updated the verbiage used throughout to indicate that Wright fishhook cactus is a globose cactus rather than a barrel cactus. Additionally, I have gone back through the introduction to reduce generalizations made about cattle impacts across cactaceae, and instead was more specific (lines 70-79). Additionally, I have incorporated multiple sources referencing the positive relationship between globose cacti and cattle (references 21-24)
- I have updated the discussion to reflect the potential for evolution of this cactus under now extinct megafauna traffic (lines 277-278).
- I have incorporated 4 additional references supporting the well documented positive response of globose cacti to moderate grazing pressures (references 21-24). Additionally, I have rewritten sections of the introduction and discussion to discuss these findings in greater detail (lines 73-79, 265-276)
- Line 320 (now 339-340) has been revised for clarity. The ethical statement was meant to state that no voucher specimens of this species were taken in order to avoid undo pressure on an endangered species.
- Have incorporated the minor edits throughout.
Thank you for taking the time to review our paper. It is apparent that you have vested interest in this subject and, as such, your feedback was extremely valuable in strengthening this work.